# Study of Micro-Samples from the Open-Air Rock Art Site of Cueva de la Vieja (Alpera, Albacete, Spain) for Assessing the Performance of a Desalination Treatment

**DOI:** 10.3390/molecules28155854

**Published:** 2023-08-03

**Authors:** Ilaria Costantini, Julene Aramendia, Nagore Prieto-Taboada, Gorka Arana, Juan Manuel Madariaga, Juan Francisco Ruiz

**Affiliations:** 1Department of Analytical Chemistry, Faculty of Science and Technology, University of the Basque Country UPV/EHU, P.O. Box 644, 48080 Bilbao, Spain; ilaria.costantini@ehu.eus (I.C.); julene.aramendia@ehu.eus (J.A.); gorka.arana@ehu.eus (G.A.); 2Department of History, Area of Prehistory, Faculty of Education Sciences and Humanities, University of Castilla-La Mancha (UCLM), Avda. de los Alfares 42, 16002 Cuenca, Spain; juanfrancisco.ruiz@uclm.es

**Keywords:** Levantine rock art, µ-Raman spectroscopy, µ-EDXRF, XRD, sulfates, biodeterioration

## Abstract

In this work, some micro-samples belonging to the open-air rock art site of *Cueva de la Vieja* (Alpera, Albacete, Spain) were analysed. These samples were collected after and before a desalination treatment was carried out, with the aim of removing a whitish layer of concretion that affected the painted panel. The diagnostic study was performed to study the conservation state of the panel, and to then confirm the effectiveness of the treatment. Micro energy dispersive X-ray fluorescence spectrometry, Raman spectroscopy, and X-ray diffraction were employed for the characterization of the degradation product as well as that of the mineral substrate and pigments. The micro-samples analysis demonstrated that the painted layer was settled on a dolomitic limestone with silicon aggregates and aluminosilicates as well as iron oxides. The whitish crust was composed by sulfate compounds such as gypsum (CaSO_4_·2H_2_O) with a minor amount of epsomite (MgSO_4_·7H_2_O). An extensive phenomenon of biological activity has been demonstrated since then in almost all of the samples that have been analysed, and the presence of calcium oxalates monohydrate (CaC_2_O_4_·H_2_O) and dehydrate (CaC_2_O_4_·2H_2_O) were found. The presence of both calcium oxalates probably favoured the conservation of the pictographs. In addition, some carotenoids pigments, scytonemin (C_36_H_20_N_2_O_4_), and astaxanthin (C_40_H_52_O_4_) were characterized both by Raman spectroscopy and by X-ray diffraction. Hematite was found as a pigment voluntarily used for the painting of the panels used in a mixture with hydroxyapatite and amorphous carbon. The results of the analyses of the samples taken after the cleaning treatment confirmed a substantial decrease in sulphate formation on the panel surface.

## 1. Introduction

Several examples of Rupestrian art, which include paintings and engravings, are still in a surprisingly preserved state. The first evidence of rock art known to this day has recently been dated with a minimum age of 45,500 years in Leang Tedongnge (Sulawesi Island, Indonesia), and this dating was based on uranium-series isotope analysis, which was conducted on two small coralloid speleothems overlying the red painting. Prior to this discovery, the first representation created was at least 43,900 years ago from an image from Leang Bulu’ Sipong 4 in the limestone karsts of Maros-Pangkep, (South Sulawesi, Indonesia) [1].

Thus, the main discoveries in terms of cave paintings occurred in South Africa [2], Argentina [3], Peru [4], Southeast Asia [5,6], Australia [7], etc., while in Europe, the most important ones were found in France and Spain, and they belonged to the transition period between the Paleolithic and the Neolithic.

Normally, this kind of artistic expression was carried out in closed spaces, such as caves or rock shelters—in other words, in cavities dug out by atmospheric agents, where populations have traditionally found shelter. These spaces suffer constant environmental impacts, making prehistoric artwork particularly fragile. This is why such spaces need constant monitoring in order to preserve them for many years to come. In this way, our research group developed a long-term monitoring methodology to better understand the conservation dynamics of rock art and its evolution over time, and this was based on the use of non-destructive elementary and molecular spectroscopies [8].

The multi-analytical approach has been widely employed in the last twenty years for the diagnostic study of the conservation state of caves and rock shelters where prehistoric art was undertaken [9]. The approach has proven to be fundamental for the study of the composition of raw materials as well as the study of painting technologies [10,11]. The palette of pigments employed was quite reduced, and was essentially composed of mineral based-pigment obtained from the natural resources of the surrounding areas [12]. Although the pigments could be used pure in most cases, a bi-colour pictograph could still be obtained, as in the case of the mixture of hematite and paracoquimbite (Fe_2_(SO_4_)_3_·9H_2_O), and this was discovered for the first time in the Abrigo Remacha rock shelter (Villaseca, Segovia, Spain) [13].

In addition, since these materials are exposed to the open air, several forms of alteration, in the form of discolorations, crusts, and patinas, which are mainly due to the impact of weathering, were characterized thank to the use of portable and laboratory analysis. As reported by Hernanz et al., a crust composed of whewellite, gypsum, calcite, clay, dolomite, α-quartz, anatase, and hematite was detected in several rock art sites in the Iberian Peninsula. Wind-blown dust and surface water runoff may also have contributed to the formation of these layers [14]. In this sense, gypsum and clayish minerals were characterized as the main components of an ochre-coloured accretion covering several parts of the third painted panel of the Hoz de Vicente rock shelter (Minglanilla, Cuenca, Spain), and were responsible for the flaking process that was observed in some areas of the painting panel [15].

In the open-air site of the engraved rock art of the Burrup Peninsula (Western Australia), high concentrations of acidic and nitrate-rich pollution, from nearby industrial complexes, provoked the colour change of the pictographs. The degradation phenomenon was due to the dissolution of manganese oxide (MnO_2_) and iron compounds, such as magnetite minerals triggered by acidic rain. This alteration induced the peeling of the rock varnish layer and produced hematite minerals, illite (K, H_3_O)(Al, Mg, Fe)_2_ (Si, Al)_4_O_10_[(OH)_2_,(H_2_O)], and kaolinite [Al_2_Si_2_O_5_(OH)_4_] [7].

The study of Pozo-Antonio et al. [16] also reported the process of colour change of rock art on a granitic outcrop at the Mougas site of Galicia (Spain). Here, the colour change phenomenon occurs on yellow and red nodes on the surface of the rock art. High temperatures provoked by wildfires cause mineral transformations (of goethite into hematite), and this increases the susceptibility of the rock to the weathering processes. Although most of the degradation phenomena are caused by atmospheric agents, degradation processes caused by anthropic factors have been discussed by Hernanz et al. [17], who detected the presence of electric welding splashes from the erection of protective iron fences around the rock art panels at the site.

In addition, gypsum and other salts, such as jarosite and bassanite, as well as biofilms, were identified in tafoni, and were generated by the weathering of sandstones (Cerro Colorado, Argentina). These secondary products were the result of impact weathering revealing hydroclastic and haloclastic processes. These activities formed active granular disintegration, and flaking and chipping affected the preservation of some painted panels [18]. Another extensive non-invasive study of Argentinean rock shelter paintings was carried out by Rousaki et al. [19]. In this study, gypsum and calcite were commonly found to be responsible for severe degradation in the form of crusts on pigmented as well as non-pigmented areas together with calcium oxalate film. Most recently, the work of Ilmi et al. [20] demonstrated that the discoloration of Leang Tedongnge (Sulawesi Island, Indonesia) rock paintings was caused by the presence of a grey/yellowish crust composed mainly of gypsum. This was caused by the reaction between the calcium ions dissolved in karst water infiltrations and the sulphate ions of minerals that are deposited on the rock surface.

The presence of biological patinas in the form of calcium oxalate monohydrate (whewellite, CaC_2_O_4_·H_2_O) and dihydrate (weddellite [CaC_2_O_4_·(2 + x)H_2_O, x ≤ 0.5]) have been frequently identified in rock shelters. They are generated by the interaction of oxalic acid, a metabolic product of microorganisms, with carbonaceous materials of the substrate [17]. Oxalate layers have also been used for attempting an indirect dating of post-Palaeolithic open-air paintings [21,22]. Although the presence of these compounds was mainly linked to the activity of microorganisms, previous studies on Ethiopian prehistoric rock painting have shown that they can also be the result of the degradation of organic matter, such as binder, that was employed in order to spread the pigments on the substrate [23]. On the other hand, the investigation of Hedges et al. [24] demonstrated the use of pigment containing calcium oxalate derived from local cacti and calcium carbonate that was probably derived from local plant ash. However, these hypotheses have not been entirely refuted, and the origin of oxalate compounds on rock paintings is a current topic that is still being investigated. On the other hand, oxalates occur naturally and are classified as organic minerals and oxalic acid in mineral deposits or in plants, fungi, and lichens, or in the form of deposits in animal tissues that are generated by diagenesis and biomineralization processes [25,26].

In the investigation of the Lower Pecos region in south-western Texas (USA), the colour change of the rock art was caused by the formation of whewellite-rich rock crust with gypsum and clay [27]. In addition, the study demonstrated a paleoclimate change from dry to wet conditions of this area, since the biopatina revealed similarities between whewellite microstructures and the desert lichen Aspicilia calcarea [28].

Therefore, due to the many factors that can degrade these extraordinary works of art, the study of both original and secondary compounds is essential in order to plan an appropriate restoration and conservation strategy for them. The current work was focused on the study of the conservation state of *Cueva de la Vieja*, which is located near Alpera (Albacete, Spain). Indeed, the main evidence of rock art in the Iberian Peninsula comes from the eastern part, and extends along the entire Mediterranean coast. The conservation site has been a UNESCO World Heritage Site since 1998 because the Levantine style used represents a unique artistic expression in the European context. Levantine art is mainly composed of paintings realized in semi-open spaces, such as rocky shelters, and it is characterized by a figurative art that is dominated by scenes of daily life and social activities, such as individual or group hunting scenes, dance, rituals, etc. Levantine art was composed mainly of paintings, and it was realized with red pigments, ochre, and oxides of manganese and iron. However, it has also been documented that there was a use of charcoal of organic origin (wood charcoal and burnt bone) as black pigments, and, moreover, a use of white earths (α-quartz, anatase, muscovite, and illite) and calcined bones (apatite) as white colours [17,29,30,31]. Numerous scientific studies have been carried out in the last 20 years with the aim of studying original compounds and degradation in order indicate the best conservation strategy [29,32,33,34].

This study is contextualised in an intervention aimed at recovering the visibility of the pictographs realized in the rock shelter, which include a diagnostic phase, the subsequent cleaning and consolidation interventions by specialized restorers, and then a final phase that will produce new digital tracing. Some samples were taken with the aim of studying the composition of the support and the white layer which covered the paintings, and which did not allow their original appearance to be appreciated from a chromatic point of view. This first approach permitted us to indicate the most adequate cleaning intervention that consisted of desalination of the panel. At the conclusion of the previous works, other samples were taken after the treatment in order to verify its effectiveness. In this work, the elemental and the molecular analyses of all of the micro-samples were performed by means of micro Raman spectroscopy, micro X-ray fluorescence, and X-ray diffraction in a laboratory.

## 2. Result and Discussion

### 2.1. Sample Analysis Prior to Cleaning Treatment

Micro X-ray fluorescence analyses were performed to define the elemental composition of the samples. For this purpose, several elemental images were acquired (Figure 1) on both sides of the sample CV01. Although the whitish layer was not appreciable by the naked eye on this sample, according to the restorers, the sulphur was homogeneously distributed throughout the exposed surface of the sample if it was compared to the inner face, where only some areas showed the presence of the element. On the other hand, calcium maps showed the presence of this element distributed throughout the piece. The presence of magnesium stood out in the CV01 sample, and it was more evident on the inner face.

In all of the micro-samples that were analysed, the presence of sulfur was less perceptible at the elemental level on the inner face, while calcium and silicon were the major elements. The main composition of the samples coming from the rocky support was based on calcium, silicon, magnesium, aluminium, iron, and potassium, with titanium varying slightly in its relative presence. On the other hand, in some samples, aggregates of titanium, zinc, chrome, manganese, copper, and chlorine were evident, which corresponded to the composition of the support. In some samples, sulfur was observed on the internal face, too, mainly in the cracking zone of the samples (see Appendix A). This suggested that the formation of sulfur compounds from the exterior part could be responsible for the phenomenon of exfoliation of the support described by the restorers.

Regarding the only micro-sample that showed traces of visible red pigmentation (Figure 2), the elemental map of iron coincided with the red area, suggesting the use of iron oxide for the painting. On the other hand, the elemental maps of this sample did not show a homogeneous layer of sulphur in the surface, as indicated by the rock-bearing analysis, and only some S hotspots were visible.

By means of Raman spectroscopy, in the majority of the points analysed on the external face of all samples, gypsum (CaSO_4_·2H_2_O, Raman bands: 180, 414, 492, 620, 670, 1008 and 1135 cm^−1^, Figure 3a) and calcium carbonate (CaCO_3_, Raman bands: 154, 282, 712 and 1986 cm^−1^) were largely detected. Even calcium magnesium carbonate dolomite (CaMg(CO_3_)_2_, Raman bands: 178, 300, 724 and 1098 cm^−1^, Figure 3c) was identified in many analysis points, and was recognized as one of the main compounds of the support. Between the original compounds of the rock, Raman spectra of anatase (TiO_2_, Raman bands 142, 395, 514, 638 cm^−1^, Appendix A), rutile (TiO_2_, Raman bands: 142, 242, 446, 612 cm^−1^, Appendix A), and quartz (SiO_2_, Raman bands 204, 264, 354, 465, 807 cm^−1^, Appendix A) were also recorded. In red and orange grains, Raman analysis showed the presence of hematite iron oxides (Fe_2_O_3_, Raman bands: 224, 245, 294, 402, 500, 612, and 1315 cm^−1^, Appendix A) and goethite (α-FeOOH, Raman bands: 204, 246, 302, 389, 480 and 550 cm^−1^), which could be responsible for the orange colour of the stone. Goethite and hematite were identified in individual grains on the specimen and, therefore, their presence does not appear to be due to the presence of pigmentation, and they are present as components of the rock.

In addition, a band at 985 cm^−1^ was detected in several spectra together with other compounds—mainly gypsum, calcite, or hematite (Figure 3b). This signal could belong to the magnesium sulfate heptahydrate so-called epsomite (MgSO_4_·7H_2_O) present on the surface of the sample as a degradation product of dolomite. Moreover, on the internal face, in most of the analysed points, the Raman spectrum of dolomite and calcium sulfate dihydrate or gypsum were observed. In several cases, the same Raman spectrum shows the coexistence of more than one compound at the same point of the analysis. In fact, several spectra have evidenced the presence of a mixture of dolomite, gypsum, and epsomite, which could clearly indicate sulfation of the original material.

This process could be favoured by the infiltration of sulfate-rich water coming from the back of the painted panel, which carries sulphates from the stone, and which accumulates them on the surface after evaporating in the open air. The formation of sulphates, especially with various hydration molecules, such as epsomite, increases the porous pressure within the rocky substrate during the hydration phase. Thus, sulfur was dissolved from rainwater, mobilized, and precipitated during the crystallisation process due to rising temperatures.

In addition, according to the archaeologists, the repeated humidification of the panel has been proven. This practice has been common to all open-air rock art sites since their discovery a couple of decades ago, and the purpose is to improve the visualization of the pictographs to the visitors. Considering that the white formation was located in the middle of the panel, where the pictographs were made, the anthropic factor could possibly be the reason for the formation of the sulphate layer. However, we cannot know exactly where the water used for this practice comes from (probably from the spring adjacent to rock shelter). It is not possible to know the composition of the water used when this practice was carried out. However, the water in the province of Albacete is characterised by a high hardness and the presence of sulphates, so much so that just this year, an osmosis plant for the treatment of drinking water was established to reduce the presence of salts and improve its quality.

The possibility of S being mobilised from the top of the rock shelter, as shown in other studies [8], up to the painted wall by a runoff process seems to be ruled out, as a percolation process from the top of the panel was not evident. On the other hand, the sulfation of the support does not appear to be due to the presence of atmospheric contaminants given that the area where the *Cueva de la Vieja* is located is not highly affected by urban traffic or industrial contamination.

With regard to the degradation by microorganisms, this was detected in both the outer and the inner parts of many of the samples. For example, in the inner face of the CV02 sample, in addition to hematite, calcite, and dolomite, the presence of a very well defined spectra of carotenoid pigments stood out. Specifically, Raman spectra of the carotenoid pigment astaxanthin (C_40_H_52_O_4_, Figure 4a), the most oxidized species among the carotenoid pigments and synthesized by species such as lichens, were recorded at several points. This identification has been made possible by the main Raman bands at 1001, 1154, and 1508 cm^−1^ and by its overtones (2150, 2298, 2509, and 2654 cm^−1^), which allowed it to be distinguished from other carotenoid pigments, such as carotene and zeaxanthin. In addition to astaxanthin, Raman spectra of scytonemin (C_36_H_20_N_2_O_4_), a pigment generally synthesized by cyanobacteria, were recorded (Figure 4b). The highest intensity Raman bands that allowed its identification were at 1170, 1382, 1554, 1600, 1632, and 1715 cm^−1^.

As observed under the microscope, the outer face of the sample CuVi03 was characterized by a homogeneous white colour. In this layer, mainly gypsum and dolomite spectra were recorded. Moreover, as observed in the image obtained with the microscope (Appendix A), the traces of hematite were clearly visible with Raman analyses. This could suggest the presence of pigmentation, as it did not appear as loose grains as it did in the previous samples, which suggests an original composition. Therefore, this would indicate the loss of polychrome. At any rate, Raman bands belonging to the calcium oxalates whewellite (192, 204, 220, 248, 895, 1461, 1488 cm^−1^) and weddelite (138, 1475 cm^−1^) were also detected in the same area. The presence of calcium oxalates could also have favoured the preservation of pigmentation in that area.

Even in the unique sample taken from a painted area, the presence of hematite was recognized by means of Raman spectroscopy as pigment that was voluntarily used. In all of the recorded spectra, the presence of calcium oxalate was also identified along with the broad bands between 1120 and 1650 cm^−1^ that belong to the aluminosilicate compounds of the substrate. Calcium oxalates were even detected in the internal face of the sample. According to previous investigations of the pictorial layer of a painted rock shelter composed mainly of hematite, it was located between layers of calcium oxalate [35], and both were caused by microorganism activity. The presence of this layer and one superior to the pictographs would be factors that have allowed the conservation of prehistoric paintings to this day. Although the presence of calcium oxalate in the internal part of the sample with red pigmentation was also identified in our samples, its small size did not allow a more in-depth study by means of a cross-sectional study of the sample.

In addition, in mixture with the iron oxide, even a single weak band located at 962 cm^−1^, typical of the hydroxyapatite (Ca_10_(PO_4_)_6_(OH)_2_) (Figure 5a), was visible in the spectra, and this suggested that the red pigment hematite was probably mixed with calcined bones. In the same spectra, moreover, broad bands belonging to amorphous carbon were also evident (Figure 5b). Thus, it is plausible that a black pigment was voluntarily added to hematite to obtain a darker colour.

Finally, molecular analysis of the crystalline part of the samples was performed with X-ray diffraction analysis. As with the other techniques, the analysis was performed on both the external and internal sides. In all samples, sample quartz, dolomite, and calcite high in magnesium were identified as original compounds, which was in agreement with the Raman spectroscopy analysis. Gypsum and compounds related to microbiological activity, such as hydrated and dihydrated calcium oxalate (whewellite and weddelite), were identified as the degradation compounds (Appendix A). Significant differences were observed between the internal and the external part. Indeed, gypsum and oxalates were detected only on the outside of the samples.

Finally, in the case of the sample containing red pigment, this technique could only identify dolomite, hydrated calcium oxalate, and gypsum as impurities. Although Raman spectroscopy identified hematite as the pigment voluntarily used, X-ray diffraction did not reveal its presence. As XRD is only sensitive to crystalline compounds, this could indicate a low crystallization of hematite in this sample. In addition, in this sample, small amounts of sulphates were also identified by all of the analytical techniques, and it is therefore plausible that the whitish sulphate layer affects the integrity of the substrate more than the areas where the pictographs are present. Undoubtedly, the presence of oxalates has guaranteed the conservation of the paintings over the years, and it is for this reason that they seem to be better preserved than the support itself. In any case, the sulfation of the panel had to be treated with the aim of bringing the pictographs back to light. In addition, the exfoliation of the support could cause the loss of the painted areas in the long term.

From this diagnostic study, it was decided that a desalination treatment by restorers should be carried out on the surface of the panel with the aim of removing the whitish layer that covered the pictographs. According to the report of the restorers, paper dressings impregnated with low mineralization water were used, and were applied directly to the sulfate crust. After a few minutes, they were removed, and then the surface was cleaned with brushes, removing the remains. The operation was repeated a couple of times, and it was completed using only distilled water.

### 2.2. Samples Analysis after the Cleaning Treatment

After completing the cleaning treatment, it was considered crucial to check its effectiveness and to verify the reduction of salts. The attention of archaeologists was also drawn to a series of points on the panel where an insoluble greyish crust remained after cleaning. These were areas arranged around natural holes in the support, through which there was, perhaps, a certain periodic emanation of water in the passage. In collaboration with the restorer who was in charge of cleaning the panel, samples were collected so that the nature of these grey crusts could be identified.

At first view, there was a considerable reduction in the surface sulphates by the naked eye. Thus, the objective of the analysis of these samples was the characterization of the rock surface after cleaning, verifying the removal of the sulfates and identifying other substances remaining in the rock in order to raise hypotheses about their link to the conservation processes in *Cueva de la Vieja*.

Although micro-EDXRF analyses still identified the presence of sulfur in the samples taken after the treatment, it was heterogeneously present in the surface of the samples. As this is a microanalysis technique, we were not surprised to detect the presence of sulphur after the cleaning treatment. However, the semi-quantitative data from this technique, based on the intensity of the emission lines, indicated a decrease in the percentage of sulphur in the samples collected after desalination. As can be seen from the micro XRF map (Figure 6), the sulfur was still present on the surface of the micro-sample due to the irregular surface, while the distribution of the calcium was homogeneous. Therefore, the treatment did not act in hard-to-reach areas.

However, from an elemental point of view, the analyses carried out on the samples after the cleaning treatment highlighted the presence of other elements whose emission bands were either weak or absent in the previously analysed samples. The presence of strontium associated with calcium was noticed in the samples after cleaning, and the bands of iron, zinc, copper, phosphorus, and manganese appeared much more evident. The emission lines of these elements belonging to the rocky support were partially hidden by the high presence of sulfur in the white layer in some of the samples.

In the three samples that were collected in grey crusts, which had not been solubilised during the cleaning treatment, high percentages of calcium were identified together with silicon to a lesser extent. Raman spectroscopy detected the presence of very sharp Raman bands of calcium carbonate that were always together with the calcium oxalate peaks (Appendix A). Under the light of the microscope, the samples looked like flat, grey-coloured flakes. No Raman spectra of gypsum were recorded in these samples. On the other hand, in one sample a strong band at 1067 cm^−1^ was visible (Appendix A). According to the literature, this band, together with the others of low intensity at 722 cm^−1^, belonged to the sodium nitrate nitratine (NaNO_3_). This compound was not identified by XRD analysis, probably because its presence was less than the detection limit of this technique. On the other hand, XRD analyses clearly identified the presence of calcium carbonate, silicon dioxide, and hydrated calcium oxalate in all three of the samples. Despite being a soluble compound, some traces of nitrate remained even after cleaning. This compound was only identified in one sample. Considering the fact that these samples were taken from an area where water runs off, the formation of nitrates must be related to traces of organic matter carried away by rainwater. In all samples, the most abundant compounds identified were calcite and calcium oxalate hydrate, which demonstrates that the treatment significantly removed the presence of sulphates without affecting the oxalate film that could contribute to the preservation of the pictographs.

On the other hand, both Raman and XRD analyses demonstrated a very low presence of gypsum in the remaining samples taken from the rock substrate. In these samples, in addition to calcium carbonate, the compounds belonging to the rocky support, dolomite, quartz, and hematite were recognized.

In order to obtain further confirmation of the effectiveness of the cleaning treatment, a statistical analysis of the data obtained by micro EDXRF was carried out.

To extract the maximum information contained in the spectra data, the use of multivariate statistical methods was commonly applied for the data treatment. The most used chemometric methods in the field of cultural heritage is the principal component analysis (PCA) applied to observe clusters of samples based on a specific variable [36]. Principal component analysis was largely employed in many investigations on several materials belonging to cultural heritage [37,38]. This statistic approach allowed the simplification of data interpretation, especially when dealing with large measurement datasets. In the literature, it was used for the identification of execution techniques, classification [39], precedence studies [40], characterization [41], correlation of secondary products and environmental agents [42,43], multispectral imaging treatment [44], colorimetric analysis [45], long-term monitoring [8], etc.

In the field of rock paintings, PCA was employed for the treatment of a data matrix of the LIBS spectra to verify the presence of clusters related to the depth profile analysis of fragments of prehistoric rock wall paintings found at two Brazilian sites. In this way, the clusters were attributed to distinct layers, and the stratigraphy of the samples was characterized [46]. In the study of Linderholm et al. [47], PCA and PLS-DA procedures were applied for near infrared spectroscopy data recorded at a Swedish Stone Age rock painting site (Flatruet, Härjedalen). Chemometric analyses of the EDXRF data was even conducted for the study of the La Peña de Candamo Cave (Asturias, Spain), which was permitted in order to distinguish different painting techniques and the most degraded areas [48].

In the field of cleaning treatment, the use of PCA is quite a new approach to evaluate its effectiveness. In the literature, in relation to cleaning procedures, PCA has mainly been applied for the treatment of colorimetric data in order to monitor natural weathering and cleaning effects on outdoor Bronze monuments patinas [49], on mosaic tiles before and after cleaning [45], or in indoor sustainable cleaning methodology for the sculpted stone of the Duomo of Milan [50].

In our study, performing Principal Component Analysis (PCA) using EDXRF spectra allowed the samples to be considerably separated considering the treatment they underwent (Figure 7). Concretely, the PC2 vs. PC3 plot made it easier to see (i.e., more explicitly) which differences appeared after applying the treatment on the samples. In Figure 7a, it can be observed how the samples before the treatment are grouped mostly on the positive part of PC2, whereas the samples analysed after the treatment are mostly grouped on the negative part of the PC2. It is clear that PC2 (Figure 7b) is the component that better explains the differences induced by the application of the conservation treatment in the samples. By analysing this PC, it can be concluded that sulfur is the main element removed during the treatment (from the elements detected by means of micro EDXRF). Actually, the samples located on the positive side of the PC2 are characterized by a high presence of sulfur, and those located on the negative side are characterized by a near absence of sulfur and a high presence of iron. This fact helps us to understand the clustering of the samples in the PC2 vs. PC3 plot, as the samples collected before the treatment are located in the more sulfur rich area and the samples collected after applying the conservation treatment appear in the region of the plot dominated by high Fe presence and low sulfur. Positive Al, Si, and Fe K bands instead characterize PC3 (Figure 7c). These elements are the main components of the rocky support, and the dispersion of the samples along this PC, especially after the treatment, is explained by the heterogeneity of natural rock. Samples before the treatment seemed to have a sulfur coating that masked rocky composition, preventing EDXRF from detecting this natural heterogeneity.

To better understand the element variations and their relationship with the studied samples, semi-quantitative XRF analyses were performed on the samples’ surface. Then, these data were studied by chemometric methods by performing scores and loading bi-plots. First of all, an outlier study was performed through the study of Hotelling T2 vs. Q residuals. After removing outliers, scores and the loading PCA bi-plot (Figure 8) showed, once again, the previously mentioned separation between the non-treated and treated samples. It is wort mentioning, in fact, that both the PCA look very similar regarding the distribution of the samples.

On the other hand, the bi-plot perfectly shows the relationship between the samples and their elemental composition. On the one hand, all of the non-treated samples are highly correlated with sulfur, as observed in the XRF spectra study. However, it must be highlighted that the B6 sample is located near the treated samples, even if it is a non-treated one. The reason for this is that this sample comes from red pictographs and, therefore, has a red coat on its surface. This red coat is composed mainly by iron shifting this sample to the area where iron dominates.

In addition, in this bi-plot, it is appreciated that the treated samples are distributed depending on the Ca and Fe content. This can aid understanding of the different hues observed in the rocky support going from whitish to reddish.

These plots also show some samples or some replicas of the same sample that instead of being non-treated are located nearer the treated samples and vice versa. This fact is again explained by the heterogeneity of a natural system such as a rock-shelter. Considering that EDXRF analysis was performed collimating the X-ray beam to 25 microns, any analyses are subjected to microscopic variation. In this way, it is logical to think that the surface of the rock walls was not perfectly treated, and that minor points remained with a high concentration of sulfur. Likewise, in non-treated samples, some microscopic points with low S presence were located. These cases are just anecdotal, though, compared with the majority of the analysed points from all of the samples (15 points for each sample were recorded).

## 3. Materials and Methods

### 3.1. Materials

The rock shelter *Cueva de la Vieja* is located to the east of the province of Albacete, 5 km away from Alpera town, and it was discovered in 1910. It was included in the UNESCO World Heritage list in 1998, as it is one of the most significant sites of the Rock Art of the Mediterranean Basin of the Iberian Peninsula. The rock shelter is a relatively deep-painted panel in full sunlight, where preserved paintings with both a large variety and a large number of figures (human figures, archers, female representations, deer, goats, bulls, horses) are distributed in a panel of 10 m length with the size of the figures ranging from 40 cm to 5 cm. All of these figures painted in red-ochre belong to the Levantine style. Even a group of abstract and geometric motifs are present that correspond to the Schematic style. The most ancient figures in the Levantine style are dated in the Epipaleolithic, which is in the transition from the Paleolithic to the Neolithic period (8000 and 6000 B.C.), while the most modern ones, in the Schematic style, belong to a period between 6000 and 3000 B.C.

The pictographs were barely visible before the cleaning treatment as a whitish layer of concretion covered most of them (Appendix A). The paintings located at both ends of the panel presented a lower incidence of this alteration and, in general, they could be observed much better. On the other hand, alterations such as runoff areas, plaques, small scales, fissures, alveolization, dissolution, and remains of animal activity were also noted. Diagnosis of these alterations and of the processes that generate them was key for verifying whether the panel would withstand cleaning of the whitish crust that makes it difficult to see the cave paintings.

Six samples were analysed before the cleaning treatment. Five of these belonged to the support, and the other one has traces of pigmentation. They were collected in degraded areas where a flaking of the support was clearly visible. After the cleaning, six other samples were collected again (Figure 9). Three of these last samples were collected from an insoluble greyish crust that remained after the cleaning procedure. These areas were located around natural holes in the support through which water could flow periodically.

### 3.2. Methods

#### 3.2.1. Micro-Energy Dispersive X-ray Fluorescence Spectroscopy (Micro-EDXRF)

The elemental maps were acquired using a M4 TORNADO EDXRF spectrometer (Bruker Nano GmbH, Berlin, Germany). The analyses were performed under vacuum (20 mbar) in order to improve the identification of the lighter elements. The lateral resolution used for spectral acquisitions was 20 μm. The maps were obtained using M-QUANT software. To obtain the quantitative maps, the assignment of the elements and the deconvolution of the spectral information were carried out. The maps were obtained by considering the K-alpha line of each element.

#### 3.2.2. Micro-Raman Spectroscopy

The micro-samples were analysed for their molecular characterization using a confocal Renishaw InVia Raman spectrometer (Renishaw plc, Wootton-under-Edge, UK) coupled to a Leica DMLM microscope (Bradford, UK). The spectra were acquired with the Leica 50× N Plan (0.75 NA) lens with a 2 μm spatial resolution. The minimum theoretical spot diameter, using the 532 nm laser, was for the Leica 50×, 0.9 μm, while using the 785 nm laser, it was 1.7 μm. Additionally, for visualization and focusing, another Leica 5× N Plan (0.12 NA) and a 20× N Plan EPI (0.40 NA) lens were used. For focusing on and searching for points of interest, the microscope implements a motorized stage (XYZ). The power applied was set at the source at a maximum of 50 mW, while on the sample, it was always less than 20 mW in order to avoid possible thermodecomposition of the samples. Normally, 10–300 scans, each lasting 1–20 s, were accumulated to achieve a suitable signal-to-noise ratio at an operating spectral resolution of 1 cm^−1^.

#### 3.2.3. X-ray Diffraction

The mineralogical composition was characterized by means of the Analytical Xpert PRO X-ray Diffractometer (XRD, PANalytical, Almelo, The Netherlands). The XRD system is equipped with a copper tube, a vertical goniometer (Bragg–Brentano geometry), a programmable divergence slit, a secondary graphite monochromator, and a Pixcel detector. The measurement conditions were set at 40 kV, 40 mA and a scan ranging between 5 and 70° 2theta. Diffractogram interpretation was performed using Win-PLOTR software by comparison with the PDF-4 standards database.

#### 3.2.4. Chemometrics

EDXRF data was analysed using chemometric methods. Two kind of analysis were performed with different aims. On the one hand, PCA analysis was performed using all the EDXRF spectra. After an outlier study, data were mean centered. On the other hand, a semi-quantitative elemental estimation was calculated from the EDXRF spectra. With this dataset and, once again, after an outlier study, the data were mean centered. By this means, a bi-plot study was performed relating scores and loadings. In this case the entire XRF spectrum was employed, while a further statistical treatment was carried out by considering the semi-quantitative data obtained with the aim of finding similarities at elemental levels between all of the samples. For PCA, we used the PLS-Toolbox v.7.0.2 (Eigenvector Research, Wenatchee, WA, USA.) implemented in MATLAB 2010 software (The MathWorks, Natick, MA, USA).

## 4. Conclusions

According to the results obtained on the micro-samples, it seems evident that the nature of the analysed stone was dolomitic limestone, with aggregates of silica or aluminosilicates as well as the presence of iron oxides, which were mainly hematite. In addition, it has been possible to identify aggregates of anatase, titanium, copper, zinc and manganese. In any case, a much-differentiated composition between the samples was not found. The only sample that presented red pigmentation demonstrated that the pictographs were conducted mainly with hematite mixed with amorphous carbon and calcined bones.

Regarding the external and internal analysis, the significant difference for the micro-samples taken before the cleaning was the presence of a whitish formation in the form of a crust on the external part. Raman and XRD analyses demonstrated that it was mainly composed of gypsum with a minor amount of epsomite. The presence of both sulfates was explained as reaction products of the original material (dolomitic limestone) as well as by the observed relative presence of both degradation compounds. Thus, it appears that a dissolution phenomenon of the original carbonate was taking place, and the following reprecipitation was as sulfates. This reactivity of the material favoured the loss of the consolidant and, therefore, its weakening. Anthropogenic factors and weathering processes contributed to the formation of sulphates on the surface of the painted panel. According to our knowledge of the history of the rock shelter, the practice of humidifiying the panel could be considered as the principal reason for the formation of sulfates, and might even be due to the distribution of the white layer that was exactly on the central area where the pictographs were made.

In addition, it was possible to observe the presence of compounds in almost all of the analysed samples that showed the presence of biological activity by the identification of calcium oxalates monohydrate and dihydrate as well as pigments such as scytonemin or astaxanthin. This indicates that microorganisms, which may also favour the dissolution of the original material and/or the acidification of the support, were affecting the material. On the other hand, it is known from previous studies that the presence of calcium oxalates is not harmful to paints, and that it would preserve their integrity over the centuries. Therefore, evaluating the dangers of the activity of microorganisms for the conservation of paintings can be a complex task.

To the naked eye, this whitish layer was removed thanks to the desalting treatment carried out by the restorers. Since microanalytical techniques have been applied to the study, they still detected the presence of sulfur after the cleaning, although in much smaller quantities. In fact, analyses performed on the samples collected after the desalination treatment showed the presence of calcium oxalates, predominantly whewellite, and showed that calcium carbonate was the main compound. This showed that the treatment removed most of the sulphate compounds without affecting the oxalate film, which is probably one of the main factors that have enabled the pictographs to be preserved.

Statistical treatment of the XRF data using principal component analysis (PCA) confirmed the effectiveness of the treatment, and it highlighted the elemental composition of each sample before and after the treatment. An increased presence of iron, manganese, zinc, copper, and potassium belonging to the rocky substrate is actually evident in the treated samples, which reveals the original nature of the panel.

Long-term monitoring will be carried out in the future by means of a portable instrument in order to check the conservation status of the panel months after the desalination treatment and to verify the influence of weathering processes on the whitish formation.

## Figures and Tables

**Figure 1 molecules-28-05854-f001:**
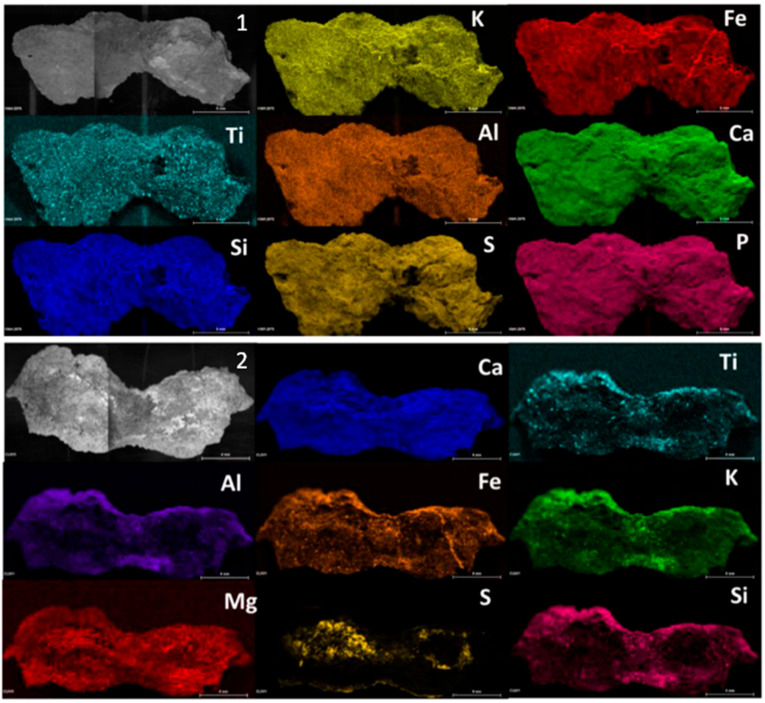
Micro-EDXRF maps of the exterior (1) and inner (2) layer, respectively, of sample CV01.

**Figure 2 molecules-28-05854-f002:**
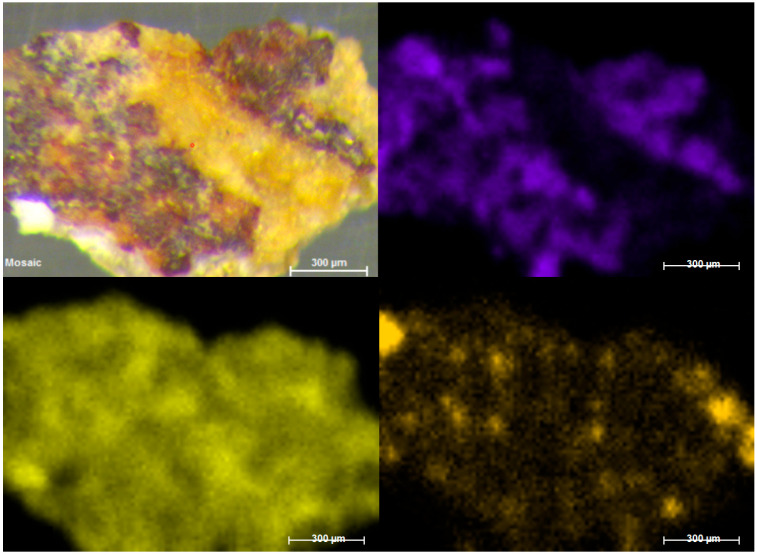
Micro-EDXRF maps of the sample with trace of red pigment.

**Figure 3 molecules-28-05854-f003:**
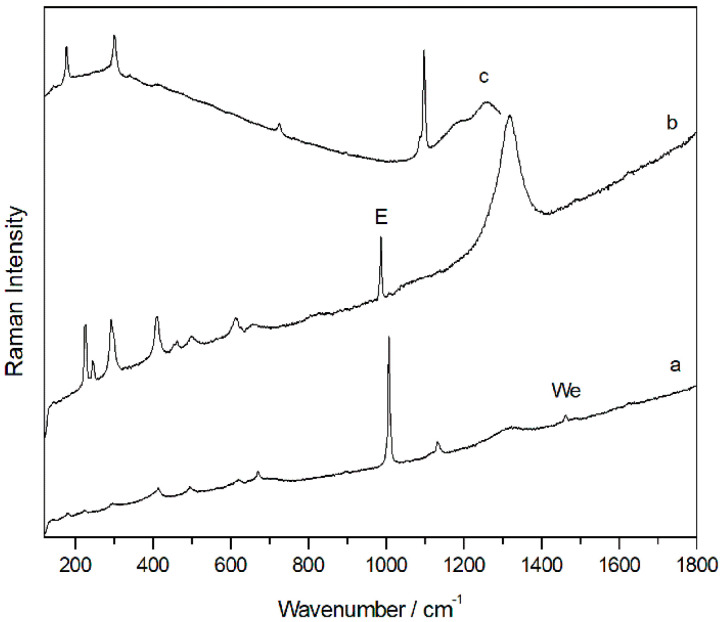
Raman spectra of gypsum with trace of weddellite (We) (**a**), hematite with epsomite (E) (**b**), and dolomite (**c**).

**Figure 4 molecules-28-05854-f004:**
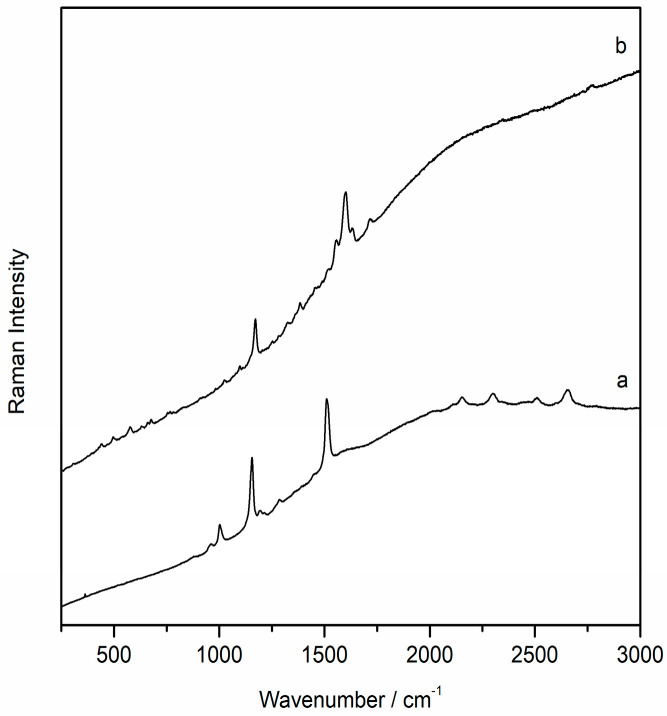
Raman spectra of astaxantin (**a**) and scytonemin (**b**).

**Figure 5 molecules-28-05854-f005:**
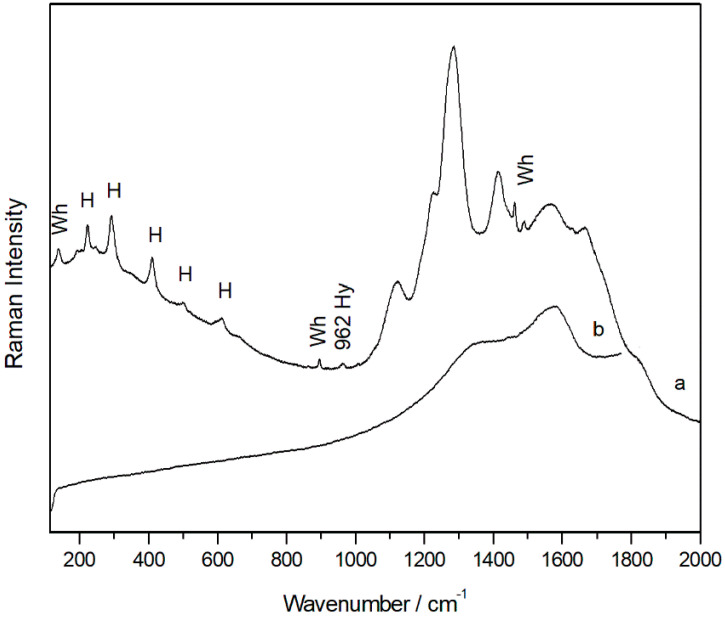
Raman spectra of hematite (H) with the signal of aluminosilicates, whewellite (Wh), and hydroxyapatite (Hy) (**a**), and black carbon (**b**).

**Figure 6 molecules-28-05854-f006:**
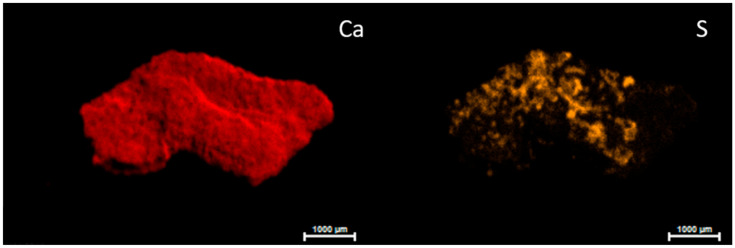
Micro-EDXRF maps of calcium and sulfur distribution after desalination treatment.

**Figure 7 molecules-28-05854-f007:**
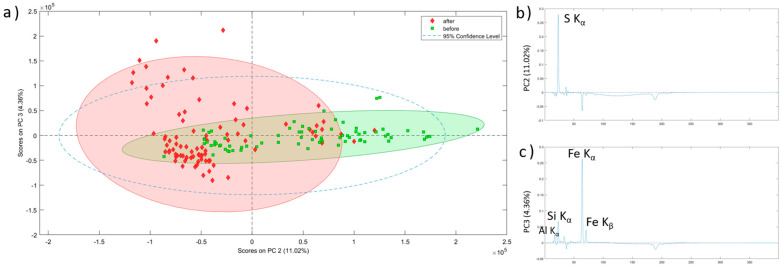
(**a**) PCA score plot of XRF spectra obtained from the sample analysis before and after the conservation treatment. (**b**) PC2 loading plot. (**c**) PC3 loading plot.

**Figure 8 molecules-28-05854-f008:**
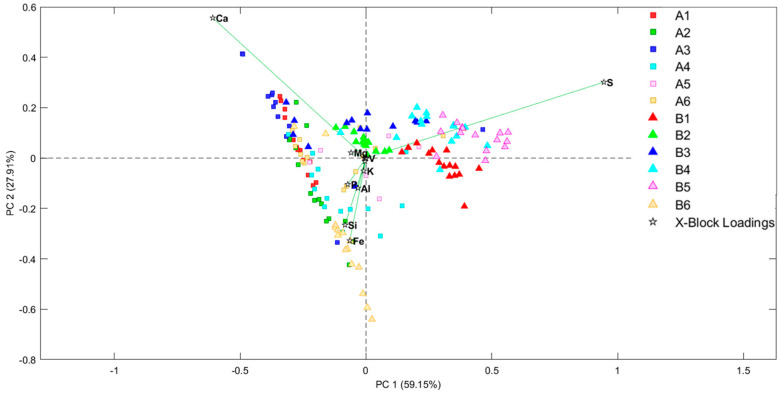
PCA bi-plot of EDXRF spectra obtained from the sample analysis before and after the conservation treatment.

**Figure 9 molecules-28-05854-f009:**
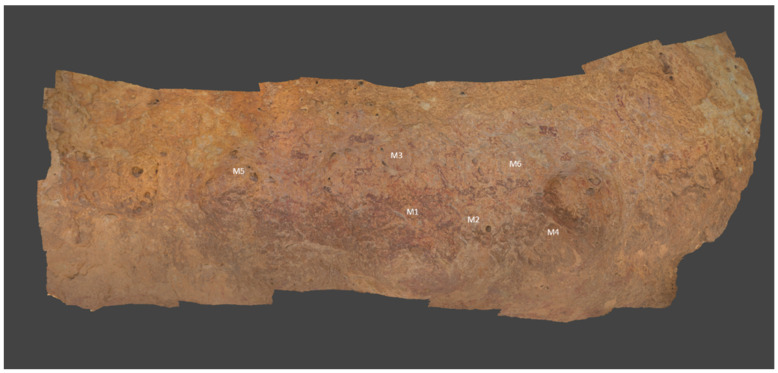
3D reconstruction de la *Cueva de la Vieja* after de cleaning treatment where the sampling points (M1–6) are marked.

## Data Availability

All data generated or analyzed during this study are included in this published article.

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
