# Peer review of "Study of Micro-Samples from the Open-Air Rock Art Site of Cueva de la Vieja (Alpera, Albacete, Spain) for Assessing the Performance of a Desalination Treatment"

_molecules, 2023, doi:10.3390/molecules28155854_

Round 1

Reviewer 1 Report

Line 14. Panel is not an appropriate term for a rock substrate.

Line 22. Check the dehydrate formula. A typo is present.

The presence of oxalates could be due to a chemical genesis, instead of a biological one. Please take into account this option

Line 80-82 unclear. Please rephrase

The oxalate film is in many ways different from a patina. Line 95 and following

Line 188-190 unclear

Line 282 please rephrase and check the language

Line 294 I have some doubt about the opportunity to name the whitish layer “patina”, because this term is part of most of terminology glossary, having  a different meaning. I suggest to maintain a sort of neutrality naming it “formation” or “layer”

Line 299 “The stability of the paint was monitored continuously during the treatment.” How? BY visual inspection?

The use of PCA to study the effect of cleaning or  desalination methods is, in my opinion, a new insight. I suggest to better introduce it, with a robust bibliography demonstrating that PCA is new in Cleaning assessment protocols

Line 420: 10 m is not a surface; it is a length

the level of english is good.

Reviewer 2 Report

This is about the analyses using the X-ray fluorescence spectrometry, Raman spectroscopy and X-ray diffraction techniques, of the micro-samples from the open-air rock art site of Cueva de la Vieja, Spain, UNESCO World Heritage included in 1998. Scientific analyses of the cultural artefacts are important and the data are meaningful by themselves. However, this manuscript contains some mistakes and requires major revision accordingly.

1.     The authors mention that 6 samples were analysed before cleaning, and other 6 samples were collected after cleaning. Can they show the locations from where the samples were taken, in Figure 9, for example?

2.     What is the relation of these six samples with the samples CuVi01, CuVi02, CuVi03, etc? Are these sample also related with the samples A1 – A6, and B1 – B6 samples in Figure 8? The relationship of these samples should be clearly described in the text.

3.     Page 3, 1st line; the 1st author name is not quoted properly; “Most recently the work of Moh. M. [20] demonstrated that …” -> “Most recently the work of Ilmi et al. [20] demonstrated that …”

4.     The sample name is “CV01” in the text line 151 of page 4, while the caption for the Figure 1 mentions sample name as “CuVi01”. Either the text or the caption should be modified accordingly.

5.     The caption for Figure 1 can be modified “Micro-EDXRF maps of the exterior (1) and inner (2) layer of the sample CuVi01 respectively.” -> “Micro-EDXRF maps of the exterior (1) and inner (2) layer, respectively, of the sample CuVi01.”

6.     It seems that the text (page 5, line 177-) describe the Figure 3, but “Figure 3” is not mentioned in the text explicitly.

7.     The caption for Figure 2. “Micro-EDXRF maps of the sample whit trace of red pigment.” -> Micro-EDXRF maps of the sample with trace of red pigment.

8.     The caption for Figure 3. “Raman spectra of gypsum with trace of weddellite (a), hematite with epsomite (E) (b) and dolomite (c)” -> Raman spectra of gypsum with trace of weddellite (We) (a), hematite with epsomite (E) (b) and dolomite (c).

9.     Line 243, page 7, “Moreover, as observed in the image obtained with the microscope (Figure), the traces of hematite were clearly visible with Raman analyses.” What is the “Figure”? It seems to mention an optical image of the sample. Where is the image?

10.  The  caption for Figure 6. “Micro-EDXRF maps where calcium and sulfur distribution after desalination treatment are showed” -> Micro-EDXRF maps of calcium and sulfur distribution after desalination treatment.

11.  Only the Figure 7 shows “A), B), C)” capital letters instead of a, b, c as in other figures. Please keep the consistency in legends. Accordingly, the caption for the Figure 7, “A) PCA score plot of XRF spectra obtained from the sample analysis before and after the conservation treatment. B) PC2 loading plot. C) PC3 loading plot” .should be changed like “PCA score plot of XRF spectra obtained from the sample analysis before and after the conservation treatment (a), PC2 loading plot (b), PC3 loading plot (c).”

12.  The legends for the x-axis and y-axis of Figure 8 are too small. The letter size for the numbers also should be increased for legibility.

13.  There are no figures corresponding to B) and C) in Figure 8. Therefore, the caption should be corrected; “PCA bi-plot of EDXRF spectra obtained from the sample analysis before and after the conservation treatment. B) PC2 loading plot. C) PC3 loading plot.” - PCA bi-plot of EDXRF spectra obtained from the sample analysis before and after the conservation treatment.

14.  Figure 9 is quoted in line 310, page 9, before Figure 6 is quoted in line 320.

15.  Supplementary figures (Figure S1, Figure S3) are also quoted, but I could not find the text that quotes Figure S2.

Some texts should be revised as commented above.

Round 2

Reviewer 2 Report

1      The caption for Figure 7 still uses “A), B), C)” capital letters instead of a, b, c. Please keep the consistency in legends. The caption for the Figure 7, “A) PCA score plot of XRF spectra obtained from the sample analysis before and after the conservation treatment. B) PC2 loading plot. C) PC3 loading plot” should be changed like “PCA score plot of XRF spectra obtained from the sample analysis before and after the conservation treatment (a), PC2 loading plot (b), PC3 loading plot (c).”

2      The legends for the x-axis and y-axis of Figure 8 are hard to read. The letter size for the numbers also should be increased for legibility.

Minor polishing required.